# Effects of Exogenous L-Asparagine on Poplar Biomass Partitioning and Root Morphology

**DOI:** 10.3390/ijms232113126

**Published:** 2022-10-28

**Authors:** Mei Han, Shizhen Wang, Liangdan Wu, Junhu Feng, Yujia Si, Xiaoning Liu, Tao Su

**Affiliations:** 1Co-Innovation Center for Sustainable Forestry in Southern China, College of Biology and the Environment, Nanjing Forestry University, Nanjing 210037, China; 2Key Laboratory of State Forestry Administration on Subtropical Forest Biodiversity Conservation, Nanjing Forestry University, Nanjing 210037, China

**Keywords:** asparagine, poplar, amino acids, protein synthesis, biostimulant

## Abstract

L-Asparagine (Asn) has been regarded as one of the most economical molecules for nitrogen (N) storage and transport in plants due to its relatively high N-to-carbon (C) ratio (2:4) and stability. Although its internal function has been addressed, the biological role of exogenous Asn in plants remains elusive. In this study, different concentrations (0.5, 1, 2, or 5 mM) of Asn were added to the N-deficient hydroponic solution for poplar ‘Nanlin895’. Morphometric analyses showed that poplar height, biomass, and photosynthesis activities were significantly promoted by Asn treatment compared with the N-free control. Moreover, the amino acid content, total N and C content, and nitrate and ammonia content were dramatically altered by Asn treatment. Moreover, exogenous Asn elicited root growth inhibition, accompanied by complex changes in the transcriptional pattern of genes and activities of enzymes associated with N and C metabolism. Combined with the plant phenotype and the physiological and biochemical indexes, our data suggest that poplar is competent to take up and utilize exogenous Asn dose-dependently. It provides valuable information and insight on how different forms of N and concentrations of Asn influence poplar root and shoot growth and function, and roles of Asn engaged in protein homeostasis regulation.

## 1. Introduction

Nitrogen (N) is a crucial element for plant growth and development, and one of the most critical limiting factors for biomass and yield formation. Hence, the efficient utilization of N is the most central event for desirable plant growth performance and productivity [1,2]. Broadly, the source of N utilized by plants in the environment can be grouped into organic and inorganic. The choice of uptake of these nutrients by plants depends on plant species and soil abundance of N (nitrate (NO_3_^−^), ammonium (NH_4_^+^), amino acids, etc.) [3], whereas the utilization of N compounds depends on the N availability and the capacity of transporters for the uptake, and the endogenous metabolism system for the subsequent utilization [4,5,6,7,8]. 

Although the main route of N uptake by plants is realized through inorganic means, the use of organic N, which can be directly absorbed by plants in the form of amino acid, has been presented by many researchers [5,9,10,11,12]. Impressive results reveal that amino acids are a very efficacious source of N for plant growth, since the majority of N in cells is bound up to them [13]. Furthermore, plants’ uptake and utilization of amino acids as a source of N are more advantageous energetically during assimilation than that of NO_3_^−^ and NH_4_^+^ [14]. Beyond their fertilizing properties, natural amino acids are nontoxic, biodegradable molecules and are potential chelators for soil remediation [15]. Moreover, amino acids represent one of the major categories of biostimulants, which have been reported to improve plant vigor, crop yield, quality characteristics, and tolerance to biotic and abiotic stresses [16]. Collectively, the interests in the application of amino acid-based N in agroforestry are increasing because of its environmentally friendly features and demonstrated capacities to improve plant growth performance [6,17,18,19,20,21,22,23,24,25,26]. Despite a long history being available about the role of amino acids in plants, their effects have varied depending on the type of amino acid over plant species; therefore, extensive studies are required to explain the physiological functions and mode of action of individual amino acids [27].

L-asparagine (Asn) is one of the twenty naturally occurring proteinogenic amino acids on Earth. It serves as a principal source of N for protein synthesis, especially in plants’ vigorously growing tissues [28,29]. Asn has carboxamide as the side chain’s functional group, which often plays a key role in the active site of enzymes [30]. Free Asn contains 21.2% N, and it has a relatively high ratio of N to C (2:4) and stability compared to the other amino acids [31,32]; thus, is commonly considered as a main organic nitrogenous storage, transport, and partitioning compound in many plant species, in particular when the supply of soluble carbohydrate is severely reduced [33]. In addition, Asn is crucial in coordinating general translation, modulating cellular signaling for amino acid homeostasis and regulating metabolic availability during biological processes [34,35]. Although Asn is a non-essential amino acid, it has been characterized as an essential exchange factor that regulates mTOR complex 1 (mTORC1, a central regulator of cell growth and division) activation, thereby coordinating protein and nucleotide biosynthesis [36]. 

Asn generally accounts for 24–76% of the xylem N in tree species [37]. The proportion of Asn in the xylem can be taken as an indicator of plant N status, wherein a decrease in N availability frequently results in a reduction in Asn and an increase in aspartate [33]. Asn is selectively accumulated in various plant tissues during normal physiological processes such as seed germination and leaf senescence, or when plants are placed in the dark [38] or exposed to a range of abiotic/biotic stresses [33,39,40,41]. The conversion of Asp to Asn in plants has been demonstrated to correlate with enhanced disease resistance [42]. Aside from the roles shown above for endogenous Asn, exogenous Asn has been found to play a part in many biological processes, such as stimulating shooting and minimizing withering of *Rosa centifolia* [13], alleviating salt stress in maize [43], altering root growth in *Arabidopsis* [44], promoting chlorophyll content, sugar metabolism and proline accumulation in cherry rootstocks [45], inducing somatic embryo formation [46], and improving *Phaseolus vulgaris* growth by stimulating cell elongation and division [47]. Furthermore, exogenous Asn has been reported as a growth regulator that positively improves vegetative growth, nutrient uptake, and pigment production in garden cress [48]. When combined with sucrose application, exogenous Asn stimulates additional outgrowth of the collateral buds in *Rosa hybrid* [49].

Poplar is a fast-growing woody plant species planted worldwide to produce wood, biomaterials, and biofuel [50,51]. Although Asn is one of the predominant translocated forms of organic N in poplar [52], the effect of Asn on poplar growth has rarely been reported so far. Previously, several studies, including ours [17,53,54], have demonstrated the ability of poplars to absorb and utilize various types of amino acids. This motivates us to conduct the present study to evaluate the effects of Asn on the vegetative growth and the physiological and biochemical traits of poplar plants.

## 2. Results

### 2.1. Exogenous Treatment of Asn Has Significant Impacts on Poplar Growth

To explore the modulation effect of exogenous Asn on poplar growth, different concentrations of Asn (i.e., 0.5, 1, 2, or 5 mM) were supplemented as a sole N source in the culture medium for poplar ‘Nanlin895’, in parallel with 0 mM and 3 mM KNO_3_ serving as the N0 and N+ control, respectively. Impressively, the morphology and the growth parameters of poplar ‘Nanlin895’ seedlings were significantly affected by Asn treatment (Figure 1 and Table 1). As seen from Figure 1, Asn treatment strongly influenced the morphophysiological traits of poplar ’Nanlin895’. Specifically, in the N0 group, plants underwent growth retardation, yellowing, and the easy falling off of mature leaves (Figure 1b). However, such effects were reversed by the treatment of varied concentrations of Asn (0.5, 1, or 2 mM) (Figure 1c–e), whose addition recorded an increase of seedling height, shoot fresh weight, leaf numbers, and leaf area by 22–65%, 63–201%, 54–86%, and 23–132%, respectively. Nonetheless, plants showed a decrease in root length (by 38–54%) and root-to-shoot ratio (by 49–62%) compared with the N0 control (Table 1). In addition, it was realized that the increment in Asn supply was inversely proportional to the plant’s growth, in that raising the concentration of the exogenous Asn to a concentration above 2 mM was observed to reduce various plant growth parameters, among which the root system in particular became thinner and darker (Figure 1f). While there were increases in fresh weights in plants treated with 0.5 and 1 mM Asn, the supply of 2 and 5 mM Asn recorded lower fresh weight than N0. Taken together, both N+ and Asn supplements had a profound impact on the morphology of the roots and shoots of poplar ‘Nanlin895’ seedlings, and the effect of Asn was dose-dependent. This is in accordance with findings reported by Haroun et al. (2010), which showed that the growth parameters of *Phaseoulus vulgaris* were induced by 1 mM Asn, but inhibited by higher concentrations of Asn (i.e., 2 mM, 3 mM, 4 mM, and 5 mM) [47]. Likewise, similar results were found by Jorkesh and Aminifard (2019) in garden cress [48].

### 2.2. Asn Affects Photosynthesis of Poplar

As earlier indicated, N strongly influences the photosynthetic rate and is an essential indicator of plants’ material accumulation [55]. In agreement with this, adding N and Asn greatly affected poplar photosynthetic and chlorophyll fluorescence parameters (Table 2). Apart from 5 mM Asn, in all other Asn treatments, there were significant increases in stomatal conductance (*g*_s_), transpiration rate (*T*_r_), and intercellular carbon dioxide concentration (*C*_i_) compared with N0. However, there was an irregular trend in the changes in the photosynthetic rate (*P*_n_), in which 1 mM Asn treatment recorded the highest *P*_n_ and 5 mM Asn recorded the lowest *P*_n_ across all the Asn treatments (Table 2). Nevertheless, the *P*_n_, *g*_s_, *C*_i_, and *T*_r_ values of poplar ‘Nanlin895’ seedlings grown under N+ conditions were remarkably higher than those of the Asn treatments.

The potential maximum light energy conversion efficiency (Fv/Fm) represents the original light energy conversion efficiency of PSII [56,57]. In the current study, there was no significant difference across all the treatments, except for the 2 mM Asn treatment, which reduced the Fv/Fm compared to N+ (Table 3). The photosystem II activity as measured by Fv/Fo depicts the potential photosynthetic capacity in the plants, and its result was consistent with the change of the Fv/Fm. The nonphotochemical quenching coefficient (NPQ) represents one of the most important mechanisms protecting plants against photoinhibition [58]. As shown in Table 3, Asn-fed poplars recorded lower values of NPQ relative to N0 and N+ groups, indicating that Asn inactivates photoprotection mechanisms. Furthermore, the photochemical quenching coefficient (qP) and the actual photochemical quantum efficiency (ΦPSII) were remarkably lower in Asn-treated plants than that of the N0 and N+ control, suggesting that extra Asn limits photosynthetic capacity. 

### 2.3. Exogenous Asn Has a Significant Influence on the Internal N Status and NUE of Poplars

The relative in vivo intensity of N and C metabolism plays an essential role in plant growth and development. To ascertain the effect of Asn treatment on poplar metabolism, the content of nitrate, ammonia, total C, total N, C/N ratio, and N use efficiency (NUE) were determined. In general, nitrate content in the shoots was higher than in the roots and was increased as Asn applied concentrations rose (Figure 2a). Moreover, in the young leaves, the treatment of Asn remarkably increased nitrate content; thus, the N+ group recorded the lowest nitrate content, followed by the N0 control. However, the opposite trends were observed in the mature leaves: the N0 group recorded the highest nitrate content, while unpredictable differences were observed in the other treatments. In the stems, the N+ control had the highest nitrate content, whereas Asn treatment reduced nitrate content to the level of the N0 control.

Nevertheless, Asn application concentration had an insignificant effect on nitrate content in the roots. Previously, it has been reported that the supply of high levels of Asn in *Pisum sativum* resulted in ammonia production [59]. In accordance with this, poplar ‘Nanlin895’-fed with Asn universally increased ammonium content in the stems, leaves, and roots (Figure 2b), indicating that exogenous Asn is indeed participating in the internal N metabolism process. 

The effect of Asn treatment over changes in C and N metabolism in the shoots and roots of poplar ‘Nanlin895’ were shown in Figure 3. In the roots, plant N accumulation (i.e., total N content) under Asn treatment increased significantly compared to N0 and N+ control. In contrast, in the shoots except for 2 mM Asn, N accumulation was insignificantly affected by other concentrations of Asn treatment compared with the N+ control (Figure 3a). Somewhat differently, C accumulation (i.e., total C content) in both shoots and roots was higher in Asn-treated plants than the N+ control (Figure 3b), implying that Asn also confers C bonus to plants. The C/N as a whole showed an opposite trend to the plant total N content (Figure 3c), wherein the highest C/N was observed in the N0 control.

The unprecedented change in plant N content suggests that Asn treatment significantly impacts poplar N metabolism. To explicit this effect, the ability of poplar to utilize N for synthesizing biomass (NUtE) and to absorb N from the soil (NUpE) under different concentrations of Asn treatment was determined. As shown in Figure 4, among all the treatments, the highest NUtE was found in 0.5 mM Asn-fed poplars, which was approximately two times higher than the N+ control. However, as exogenous Asn concentration increased, NUtE dropped sharply. Treatment with 1 mM Asn and 2 mM Asn reduced NUtE to 10.4 and 3.8 g dry weight per gram N, respectively (Figure 4a). The changing trend of NUpE was similar to that of NUtE in poplar ‘Nanlin895’ seedling, in which treatment with 0.5 mM Asn recorded two times higher NUpE than the N+, whereas NUpE of the rest of the Asn treatment fell below the level of the N0 control (Figure 4b). Taken together, different concentrations of exogenous Asn significantly influenced poplar NUE, among which 0.5 mM Asn treatment recorded the highest NUE, whereas ascending concentration of Asn in poplar ‘Nanlin895’ relatively decreased NUE. Consistently, it has been demonstrated that higher NUE is observed in plants with low N supply [60].

### 2.4. Effects of Exogenous Asn on the Contents of Amino Acids

The impact of exogenous Asn as a sole N source on the endogenous amino acids content in both roots and mature leaves of poplar ‘Nanlin895’ was shown in Figure 5. Generally, high amounts of Asn, arginine, glutamic acid, alanine, and aspartic acid but low contents of phosphoserine, glycine, methionine, and tyrosine were detected in all the samples. Varying the concentration of exogenous Asn dramatically affected the range of free amino acids in both roots and mature leaves of poplar ‘Nanlin895’. In brief, the content of the amino acids in the roots was only marginally affected by 0.5 mM Asn, but further increasing exogenous Asn dose augmented the content of a number of amino acids, especially the internal Asn, arginine, GABA, and alanine compared with the N+ control. Intriguingly, 5 mM Asn treatment led to a substantial accumulation of almost all amino acids, while the N0 control recorded their lowest level (Figure 5a). Overall, the application of Asn had similar effects on the amino acid content in the mature leaves compared to the roots, except that in the mature leaves the N+ control recorded the lowest level of all amino acids, while 0.5 mM Asn treatment enriched the contents of a proportion of the amino acids examined (Figure 5b). 

### 2.5. Impacts of Asn Treatment on the Expression of Genes Involved in N and C Metabolism Pathways

The root system architecture is vital in regulating root foraging response to N fluctuation [61]. As aforementioned, the application of Asn significantly influenced poplar ‘Nanlin895’ root growth. To explore the underlying molecular basis, quantitative real-time RT-PCR (qPCR) was implemented to determine the expression levels of key genes involved in N and C metabolism pathways in the roots of poplar ‘Nanlin895’ fed with different concentrations of Asn. The results (Figure 6) showed substantial individual variability within different gene isoforms. In higher plants, glutamine synthetase (GS), the key enzyme engaged in N assimilation, is composed of two isoenzymes: the cytoplasmic GS1 (contained GS1.1, GS1.2, and GS1.3) and the chloroplast GS2 (encoded by a single gene *GS2*) [31]. Asn treatment significantly induced the expression level of *GS1.1* and *GS2* compared with both N0 and N+ control; however, the induction effect was negatively correlated with the Asn concentration applied. Similarly, the expression of *GS1.3* was slightly induced by Asn except for a tentative depression by 2 mM Asn treatment. On the contrary, the expression level of *GS1.2* was strongly inhibited by external Asn. Glutamate synthase (GOGAT) is another key enzyme involved in inorganic N assimilation. It has two forms in plants: ferredoxin-dependent (Fd-GOGAT) and NADH-dependent (NADH-GOGAT). Asn treatment concentration had a marked impact on the expression of both forms. While 0.5 mM Asn treatment dramatically increased *Fd-GOGAT* and *NADH-GOGAT* transcripts, 5 mM Asn treatment notably decreased them. Asparagine synthetase (AS) and asparaginase (ASPG) are the prime enzymes in plants participating in Asn synthesis and degradation, respectively [28]. The transcripts of *AS1* and *AS3* were positively induced by the external Asn, whereas the transcripts of *AS2* were selectively induced by 0.5 mM Asn and 2 mM Asn (to a less extent) compared with the N+ control (Figure 6). Conversely, the expression of *ASPG2* was repressed by 0.5 mM Asn but augmented when Asn concentration rose. Nevertheless, the expression levels of *ASPG1* and *ASPG3* were predominantly stimulated by 0.5 mM Asn but moderately changed among other Asn treatments, except for a decline under 2 mM Asn treatment. Likewise, *nitrate reductase* (*NR*) expression was significantly increased by different concentrations of Asn apart from 1 mM Asn. The *nitrite reductase* (*N**iR*) transcriptional level was slightly influenced by Asn treatment compared with the N+ control. The cell wall invertase (CWINV) and the vacuole invertase (VI) are two sucrose breakdown enzymes important for primary carbon metabolism [62]. In contrast to the above N-related genes, the transcripts of *CWINV1* were enhanced only by 0.5 mM Asn but suppressed by other concentrations of Asn treatment, while the transcripts of *VI2* were strikingly induced by Asn treatment.

### 2.6. Impacts of Asn Treatment on Enzymes Involved in N Metabolism

The enzymes involved in N metabolism pathways play fundamental roles during plant growth and development. To elucidate the biochemical effects of exogenous Asn at the enzyme level on poplar ‘Nanlin895’, the activities of key enzymes including GS, GOGAT, Aspartate aminotransferase (AspAT), and glutamate dehydrogenase (GDH) involved in N metabolism were analyzed in poplar roots. The AS and ASPG were omitted because reliable measurement of their enzyme activities was recalcitrant due to the presence of natural inhibitors and cross-reactions [63]. The enzyme activity assay results (Figure 7) showed that the application of Asn as a sole N source substantially altered the activities of the above enzymes. However, their regulatory responses varied with the change in Asn concentrations. While 0.5 mM Asn treatment significantly increased the activities of all four enzymes in poplar roots, 5 mM Asn treatment decreased their activities, though not to a marked degree compared with the N+ control. Among all of the four enzymes, GOGAT activity was most sensitive to Asn treatment, which was increased by more than 10 times (1 mM Asn treatment) than the N+ control, and its changing trend was consistent with the change in corresponding transcripts in 0.5 mM and 5 mM Asn-treated plants. In contrast, GS activity was less affected by Asn treatment, although strong inductions of *GS1.1*, *GS1.3,* and *GS2* mRNA expression by 0.5 mM Asn and inhibition of *GS2* by 5 mM Asn treatment were observed. The change in GDH activity was inversely correlated with Asn concentration, and the highest activity was found in 0.5 mM Asn-treated poplars, while the lowest activity was found in 5 mM Asn-treated poplars. Asn treatment increased AspAT activity; however, the change in AspAT activity was somewhat irrelevant to Asn treatment concentration.

## 3. Discussion

Plant growth is correlated with increased synthetic demands, which require a constant supply of amino acids to support protein synthesis and matrix production. Amino acids are environmentally friendly molecules, and their uptake and utilization as an N source by plants is energetically more advantageous than inorganic N (including NO_3_^−^ and NH_4_^+^ or N_2_) since it circumvents the mineralization pathway [14]. However, different types and concentrations of amino acids have a huge difference in their intake and utilization efficiency [17,54], which makes the effects of amino acids on plant growth vary. Asn serves as a principal amino acid for protein synthesis, transporting, and storing N in plant cells [64]. Asn-C is enriched in aspartate, malate, fumarate, and citrate, and Asn-N is enriched in aspartate, glutamate, proline, serine, and alanine [65]. Earlier works conducted in *Arabidopsis* and lettuce showed impressive growth results when Asn was used as a sole N source [66,67]. Moreover, the supply of Asn resulted in an improvement in relative water content, an increase in photosynthetic pigments, and a decrease in electrolyte leakage from cellular membranes, as well as a reduction in leaf proline content and activities of key oxidative relevant enzymes in salt-stressed maize plants [43]. Our prior study showed that exogenous glutamine as a single N source could promote poplar growth [53]. Given that Asn and glutamine are structurally similar since they both contain amide groups in their respective side chains, we therefore applied different concentration of Asn to poplar ‘Nanlin895’ to explore its effects on poplar biomass and its potential mode of action. In general, our study uncovers the nutritional effects and biochemical basis of exogenous Asn on poplars. It provides experimental evidence on the capacity of Asn to support poplar growth. Specifically, careful observation and comparison of plant phenotypes and physiological and biochemical analysis revealed that poplar ‘Nanlin895’ grew sturdily in Asn culture solution, which indirectly confirmed that Asn was indeed taken up and utilized by poplar. That notwithstanding, when Asn was added to the N-free nutrient solution, the N deficiency stress impacts on poplar (seen in N0 control) were relieved. Among all the concentrations of Asn applied, poplar ‘Nanlin895’ grew best at 0.5 mM Asn, as reflected by the seedlings’ height, biomass, and photosynthetic rate. Adding 1 mM Asn did not further improve poplar growth, but resulted in a low photosynthetic rate and a low conversion rate of maximized potential light energy. When exogenous Asn concentration was increased to the level of above 2 mM, it caused a high N stress response and the seedlings’ growth rate began to drop. This resultantly led to shortness in height, yellow and black roots, and a reduction in biomass and photosynthetic rate of poplar ‘Nanlin895’ seedlings. In conclusion, physiological reactions of poplar ‘Nanlin895’ seedlings to different levels of Asn were divergent based on parameters examined. Our study corroborates the dose-dependent effects of Asn on plants’ growth [47,48].

NUE basically depends on plants’ efficient uptake and use of N to increase biomass [68,69]. Earlier on, it had been demonstrated that the application of phenylalanine improved NUE in *Populus* X *canescens* [17], and the supply of glutamine (0.5 mM) increased NUE in poplar ‘Nanlin895’ [53]. Accordingly, we found that poplar ‘Nanlin895’ feeding with 0.5 mM Asn had higher NUtE and NUpE than the N+ control. Nevertheless, increased concentration of Asn was inversely related to the NUtE and NUpE ascribed to N wastage, which led to the overreading of N input and a decline in N utilization. By contrast, adding 0.5 mM Asn may sustain the optimum N/C ratio (Figure 3) and amino acid homeostasis for the effective absorption, utilization, and efficient N transformation within plants. Collectively, under the current experiment conditions, 0.5 mM Asn was the optimal treatment concentration for poplar ‘Nanlin895’ growth. Although Asn application is not essential for poplar, our results demonstrate that the combination of fertilization and biostimulation effect of Asn can be aligned with the effective use of conventional inorganic N.

It has been known that N content significantly affects the level of free amino acids in plants [70,71]. The feeding of Asn [U-^14^C] to rice seedlings resulted in the predominant labeling of glutamine, Asn, and glutamic acid in the shoots, as well as Asn and γ-Aminobutyric acid (GABA) in the roots [72]. Accordingly, in our study, the application of Asn as a sole N source in the hydroponic nutrient solution dramatically influenced the content of asparatic acid, glutamate, alanine, GABA, arginine, and endogenous Asn in both roots and mature leaves of poplar ‘Nanlin895’ (Figure 5). In particular, the contents of the amino acids Asn, arginine, and GABA, which are involved in environmental response [41,73,74], were profoundly enhanced in poplar, showing a positive changing trend in proportion to Asn application dose. This result indicates that the exogenous application of Asn is actually utilized by poplar, and it promotes the transformation of amino acids within the plant. However, a high negative correlation between amino acid accumulation (Figure 5) and root growth (i.e., root biomass and length) (Figure 1), along with an increase in Asn applied dose, was found. Meanwhile, it was also observed that the enzymes associated with N metabolism—GS, GOGAT, GDH, and AspAT—were increased by 0.5 mM Asn treatment but declined by 5 mM Asn. Previously, it was reported that the primary use of Asn in mammalian cells is in protein synthesis [75]. Consistently, a [^14^C] labeling study found that Asn was used for protein synthesis on a quantitatively significant scale in plants [76]. Given that the presence of Asn in large excess is a good marker of protein synthesis and breakdown [77], it is therefore reasonable to assume that the accumulation of amino acids observed in our study in poplar roots in response to high concentration of exogenous Asn (5 mM) is relevant to protein catabolism. In agreement with this possibility, it has been shown that Asn accumulation suppresses the well-studied eukaryotic translation initiation factor-2 alpha (eIF2α) kinase GCN (general control nonderepressible) and reduces the activating transcription factor 4 (ATF4), thereby impeding protein synthesis and cell growth, whereas Asn dissipation activates GCN and increases ATF4, and hence supports the translation of protein and maintains cell growth [75]. 

Overall, in our study, the root morphology of poplar ‘Nanlin895’ was greatly affected by Asn, implying that the root was the main site where Asn regulation took effect under the current experimental conditions. Our results indicate that Asn may function not only as a metabolism but also as a signaling/regulatory candidate as a part of the plant adaptation process, activating multiple responses related to N assimilation [39]. The possible mode of action of Asn is illustrated in Figure 8. It can be proposed that under high doses of Asn treatment, protein degradation rather than protein synthesis occurs in poplar ‘Nanlin895’; that is, an overdose of Asn (5 mM) disrupts protein synthesis probably through the GCN2/ATF4 pathway, followed by amino acid accumulation, a reduction in enzyme activity of GDH and GOGAT, and in turn, poplar growth (root length and biomass in particular) depression, whereas the application of an appropriate level of Asn (i.e., 0.5 mM) may facilitate nascent protein synthesis as a result of poplar growth promotion. As the transcriptional change of GCN2/ATF4 pathway genes was only marginal by qPCR analysis, we deduce that Asn regulation occurs at the post-translation level, and phosphorylation of eukaryotic translation initiation factors is likely engaged in this process [75,76,78].

## 4. Materials and Methods

### 4.1. Plant Material and Growth Conditions

Plantlets of poplar ‘Nanlin895’ (*Populus deltoides* × *P. euramericana*) clones that had been tissue-cultured in 1/2 MS medium for one month were used throughout the experiment, unless otherwise indicated. All seedlings were initially kept in pure water to get rid of internal N for 10 days, then treated with inorganic-N (KNO_3_) or various concentrations of Asn-N in the N-free 1/2 MS nutrient solution. There were six groups of treatments in total: the standard inorganic-N control (3 mM KNO_3_; N+), the N-free control (0 mM N; N0), 0.5 mM Asn, 1 mM Asn, 2 mM Asn, and 5 mM Asn. The nutrient solution was refreshed twice a week for two months. Poplar seedlings were kept in disposable plastic water cups for the hydroponic treatment, one seedling per cup. Each cup contained 250 mL nutrient solution fixed with a black foam board. Each treatment included at least six individual plants. The whole experiment was repeated three times. The seedlings were grown under 24 °C light/22 °C dark, 5000 Lux light intensity for 16 hours with 40–60% humidity. At the end of the treatment, tissues from young leaves (YL), mature leaves (ML), stems (S), and roots (R) were collected, snap-frozen in liquid nitrogen, and stored at −80 °C for further analysis.

### 4.2. Analysis of Growth, Gas Exchange, and Chlorophyll Fluorescence Characteristics

Growth parameters including plant height, root length, and fresh shoot weight, as well as fresh root weight, were measured before harvest. LI-6400 Portable photosynthesis system (LI-COR, Lincoln, NE, USA) was used to measure the net photosynthetic rate (*P*_n_, μmol CO_2_·m^−2^·s^−1^), intercellular carbon dioxide concentration (*C*_i_, μmol·mol^−1^), stomatal conductance (*G*_s_, mol H_2_O·m^−2^·s^−1^), and transpiration rate (*T*_r_, mmol H_2_O·m^−2^·s^−1^). During the measurement, environmental conditions were set at 800 μmol·m^−2^·s^−1^ of light intensity and 400 μL·L^−1^ of carbon dioxide. The nonphotochemical quenching coefficient (NPQ), photochemical quenching coefficient (qP), the actual photochemical quantum efficiency of PSII (ΦPSII), the potential maximum light energy conversion efficiency (Fv/Fm), and photosystem II activity (Fv/Fo) were measured using chlorophyll fluorescence imaging system (CF Imager, Nanjing, China).

### 4.3. Determination of Activities of Enzymes Involved in N Metabolism

The activities of key enzymes—including nitrate reductase (NR, EC 1.6.6.1), glutamine synthetase (GS, EC 6.3.1.2), glutamate synthase (GOGAT, EC 1.4.7.1), aspartate aminotransferase (AspAT, EC 2.6.1.1), and glutamate dehydrogenase (GDH, EC 1.4.1.2)—that are involved in plant N metabolism pathways [68] were analyzed in poplar ‘Nanlin895’ leaves, stems, and shoots following methods described in earlier studies [69,79,80].

### 4.4. Measurement of the Content of Free Amino Acids, NH_4_^+^, NO_3_^−^ and NUE

The free amino acids were extracted by sulfosalicylic acid according to previous methods [81], and the content of free amino acids was determined by the automatic amino acid analyzer (S-433D, Sykam GmbH, Eresing, Germany) following the operation instruction. The ammonium and nitrate content was determined by the modified ninhydrin colorimetric criteria [82] and Patterson and colleagues’ method [83], respectively. Nitrogen use efficiency was calculated according to previously reported equation: NUtE = biomass DW/N supply, NUpE = total nitrogen content/N supply [84].

### 4.5. Analysis of Transcript Levels of Genes Involved in N Metabolism

Quantitative RT-PCR was employed to analyze the expression levels of nitrate reductase (NR), nitrite reductase (NiR), asparagine synthetase (AS), asparaginase (ASPG), glutamine synthase (GS), glutamate synthase (GOGAT), cell wall apoplastic invertase (CWINV1), and vacuolar invertase (VI2) genes in the leaves, stem, and roots of poplar. Total RNA was extracted by using RNAprep Pure Polysaccharide Polyphenol Plant Total RNA Extraction Kit (TIANGEN, Beijing, China). The purity and concentration of the extracted RNA were checked by the Microvolume Spectrophotometer (Colibri LB 915, Bad Wildbad, Germany) and then verified through agarose gel electrophoresis. Reverse transcription kit (PrimeScriptTM RT reagent Kit with gDNA Eraser, Takara, Japan) was used to synthesize the first strand of cDNA. Gene-specific primers (Appendix A) were designed by Primer3 (http://primer3.ut.ee/, accessed on 21 February 2022). Quantification of gene amplification was performed on the real-time fluorescent quantitative PCR instrument (Applied Biosystems StepOne^TM^, Beijing, China) with the fluorescent agent AG SYBR Green Pro Taq HS premixed qPCR kit (Accurate Biology, Changsha, China).

### 4.6. Statistical Analysis of Data

Analysis of variance (ANOVA) was performed for all data using IBM SPSS Statistics 21 statistical software. The standard errors were based on the pooled error term from the ANOVA table. Differences were considered statistically significant when the *p*-value of the ANOVA F-test was less than 0.05.

## 5. Conclusions

Asn is a multipurpose amino acid participating in various biological processes, including plant development and adaptation to environmental stresses. Our study found that the application of exogenous Asn significantly increased the biomass of poplar ‘Nanlin895’ compared to the N-free control by improving the growth status of N starvation. To the best of our knowledge, this is the first research of exogenous application of Asn as a sole N source conducted on poplars. It reveals that Asn, when supplied at an appropriate rate, is conducive to the growth fitness of poplar. Although there was no match for the growth promotion effect of Asn as against the N+ control, it was, however, clear that relative to the N-free plants, application of Asn dramatically alleviated poplar ‘Nanlin895’ seedlings from N deficiency stress. In consistency with previous findings, it can be concluded that the concentrations and forms of N influence poplar growth and performance, including biomass partitioning and root morphology such as root length and root biomass [85]. 

Our research provides essential information on Asn for effective biomass allocation in poplars. It is reasonable to infer that (1) poplars are competent at using Asn as a sole N source to support their growth, as indicated by the growth parameters and photosynthesis activities of poplar ‘Nanlin895’ plantlets fed with Asn; (2) the feeding of Asn at a proper concentration leads to root morphology alteration, as a result of increasing/decreasing nutrient and possibly water absorption; (3) the exogenous application of Asn induces the transcript levels of key genes and activity of enzymes involved in N and C metabolism pathway, promotes ammonium assimilation, mitigates N metabolic disturbance resulting from N stress, and advances N absorption and utilization in both roots and the above-ground tissues; (4) Asn might play a critical role in regulating protein homeostasis, and in turn, plant growth and stress response.

## Figures and Tables

**Figure 1 ijms-23-13126-f001:**
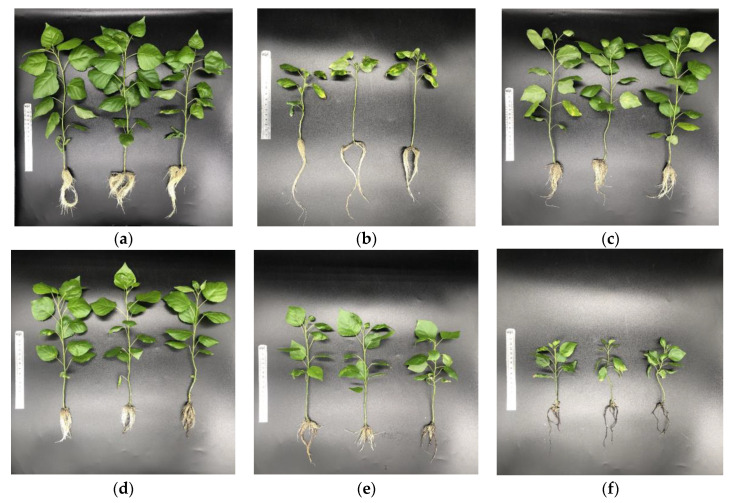
Phenotypical changes of exogenous asparagine (Asn) on poplar ‘Nanlin895’ growth. Plants were grown in (**a**) 3 mM KNO_3_ (N+), (**b**) N-free hydroponic solutions (N0), (**c**) 0.1 mM Asn, (**d**) 0.5 mM Asn, (**e**) 2 mM Asn, and (**f**) 5 mM Asn for two months.

**Figure 2 ijms-23-13126-f002:**
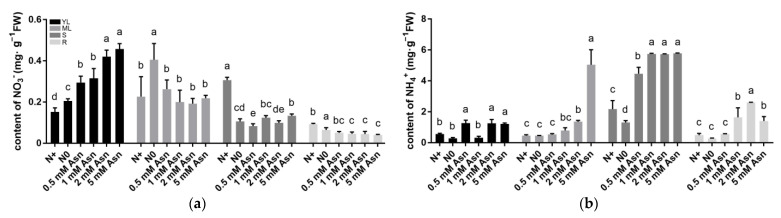
The contents of nitrate (**a**) and ammonia (**b**) in various tissues of the poplar ‘Nanlin895’ under different asparagine (Asn) treatments. “N0” indicates N-free; “N+” represents the inorganic N control. The concentration of Asn applied was 0.5 mM, 1 mM, 2 mM, or 5 mM, as indicated. YL, young leaves; ML, mature leaves; S, stems; R, roots. Bars indicate means ± SD (n ≥ 3), and different letters indicate significant differences (*p* < 0.05) based on Duncan’s test.

**Figure 3 ijms-23-13126-f003:**
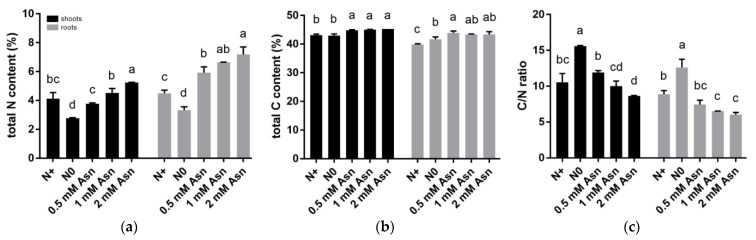
Effects of exogenous asparagine (Asn) on the nitrogen (N) and carbon (C) contents in the shoots and roots of poplar ‘Nanlin895’. (**a**) total N content; (**b**) total C content; (**c**) the C/N ratio. “N0” indicates N-free reference; “N+” represents the inorganic N control. The concentration of Asn applied was 0.5 mM, 1 mM, and 2 mM as indicated. Bars denote means ± SD (n ≥ 3), and different letters indicate significant differences (*p* < 0.05) based on Duncan’s test.

**Figure 4 ijms-23-13126-f004:**
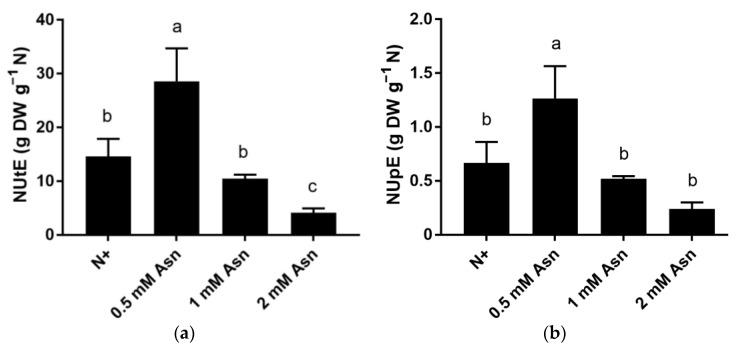
Nitrogen utilization efficiency (NUtE) (**a**) and nitrogen uptake efficiency (NUpE) (**b**) of poplar ‘Nanlin895’ under different asparagine (Asn) treatment. “N0” indicates N-free; “N+” represents the inorganic N control. The concentration of Asn applied was 0.5 mM, 1 mM, and 2 mM as indicated. Bars denote means ± SD (n ≥ 3), and different letters indicate significant differences (*p* < 0.05) based on Duncan’s test.

**Figure 5 ijms-23-13126-f005:**
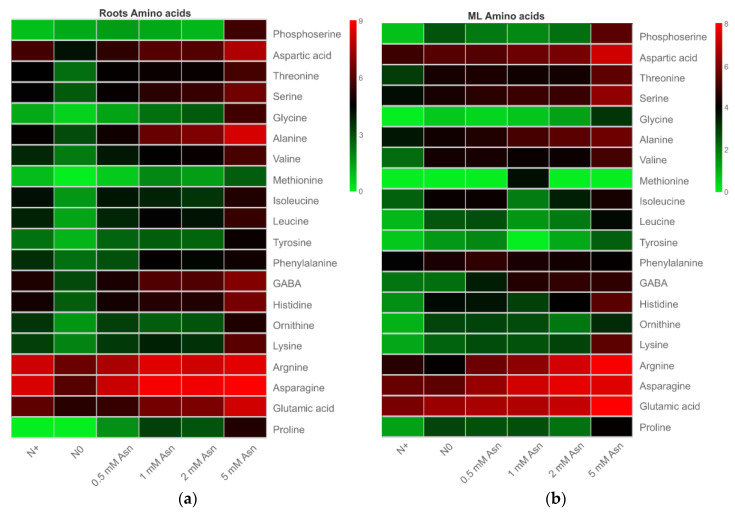
Heatmap illustration of the amino acid composition of poplar ‘Nanlin895’ roots (**a**) and mature leaves (**b**) under different asparagine (Asn) treatments. “N0” indicates N-free; “N+” represents inorganic N control. The concentration of Asn applied was 0.5 mM, 1 mM, 2 mM, and 5 mM as indicated. The color bar indicates low (green) expression levels to high (red).

**Figure 6 ijms-23-13126-f006:**
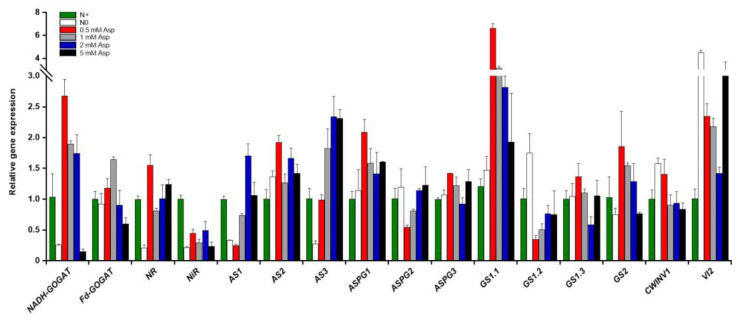
Transcriptional changes of genes related to the N and C metabolism pathways in poplar ‘Nanlin895’ roots in response to asparagine (Asn) treatment. *NR*, nitrate reductase; *GS*, glutamine synthase; *NADH-GOGAT*, nicotinamide adenine dinucleotide-dependent glutamate synthase; *Fd-GOGAT*, ferredoxin-dependent glutamate synthase; *AS*, asparagine synthetase; *ASPG*, asparaginase; *N**iR*, nitrite reductase; *CWINV*, cell wall apoplastic invertase; *VI*, vacuolar invertase. “N0” indicates N-free reference; “N+” represents the inorganic N control. The concentration of Asn applied was 0.5 mM, 1 mM, 2 mM, and 5 mM as indicated.

**Figure 7 ijms-23-13126-f007:**
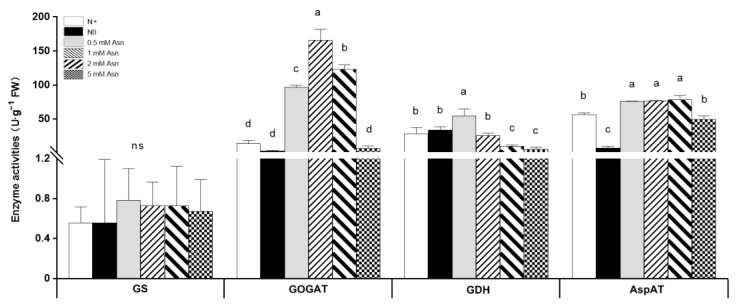
Effects of exogenous asparagine (Asn) on the activities of enzymes involved in N metabolism pathways in poplar ‘Nanlin895’ roots. GS, glutamine synthase; GOGAT, glutamate synthase; GDH, glutamate dehydrogenase; AspAT, aspartate aminotransferase. “N0” indicates N-free; “N+” represents the inorganic N control. The concentration of Asn applied was 0.5 mM, 1 mM, 2 mM, and 5 mM. Bars indicate means ± SD (n ≥ 3), and different letters indicate significant differences (*p* < 0.05) based on Duncan’s test.

**Figure 8 ijms-23-13126-f008:**
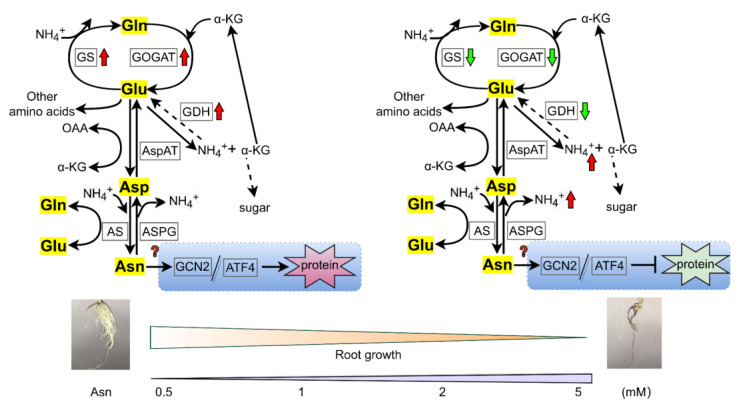
Schematic model of Asn dose-dependent effects on poplar roots. Gln, glutamine; Glu, glutamic acid; Asp, aspartic acid; α-KG, α-ketoglutarate. GS, glutamine synthase; GOGAT, glutamate synthase; AS, asparagine synthetase; ASPG, asparaginase; GDH, glutamate dehydrogenase; AspAT, aspartate aminotransferase; GCN, general control nonderepressible; ATF, activating transcription factor. The up arrow indicates induction and the down arrow indicates depression. The box stands for the enzyme.

**Table 1 ijms-23-13126-t001:** Physiological effects of exogenous asparagine (Asn) on poplar ‘Nanlin895’ growth.

Treatment	Height (cm)	Root Length (cm)	Shoot FW (g)	Root FW (g)	Number of Blades	Leaf Area (mm^2^)	Root–Shoot Ratio
N+	30.767 ± 2.926 ^a^	16.467 ± 4.563 ^a^	6.115 ± 1.589 ^a^	2.068 ± 0.356 ^a^	14.333 ± 0.577 ^a^	1562.667 ± 483.514 ^a^	0.348 ± 0.083 ^b^
N0	16.275 ± 2.837 ^de^	17.7 ± 3.403 ^a^	1.301 ± 0.599 ^e^	0.845 ± 0.298 ^b^	7 ± 0.816 ^d^	723.267 ± 429.444 ^b^	0.671 ± 0.076 ^a^
0.5 mM Asn	26.8 ± 2.342 ^b^	10.975 ± 3.2 ^b^	3.916 ± 1.236 ^b^	0.943 ± 0.208 ^b^	13 ± 1.826 ^ab^	1676.667 ± 529.802 ^a^	0.255 ± 0.08 ^b^
1 mM Asn	23.333 ± 2.307 ^bc^	9.5 ± 0.5 ^b^	3.016 ± 0.259 ^bc^	0.881 ± 0.052 ^b^	11.333 ± 1.155 ^bc^	1018.5 ± 105.194 ^ab^	0.293 ± 0.022 ^b^
2 mM Asn	19.85 ± 2.029 ^cd^	8.075 ± 2.329 ^b^	2.12 ± 0.345 ^cd^	0.723 ± 0.165 ^b^	10.75 ± 0.5 ^c^	886 ± 542.769 ^ab^	0.339 ± 0.025 ^b^
5 mM Asn	12.533 ± 0.551 ^e^	11.533 ± 1.858 ^b^	1.023 ± 0.207 ^e^	0.319 ± 0.042 ^c^	8.667 ± 0.577 ^d^	511.4 ± 237.216 ^b^	0.32 ± 0.076 ^b^

Physiological parameters were measured in poplar under different asparagine (Asn) treatments. “N+” represents inorganic N control; “N0” indicates N-free control. Data indicate means ± SD (n ≥ 3). Different letters behind the values in the same column indicate significant differences between the treatments, tested by one-way ANOVA followed by Dunken’s test (*p* < 0.05).

**Table 2 ijms-23-13126-t002:** Photosynthetic parameters of the poplar ‘Nanlin895’ under different asparagine (Asn) concentrations.

Treatment	*P*_n_ (μmol CO_2_·m^−2^·s^−1^)	*g*_s_ (mol·m^−2^·s^−1^)	*C*_i_ (μmol·mol^−1^)	*T*_r_ (mmol H_2_O·m^−2^·s^−1^)
N+	8.095 ± 0.533 ^a^	0.152 ± 0.033 ^a^	320.2 ± 14.306 ^a^	1.468 ± 0.254 ^a^
N0	2.318 ± 0.299 ^f^	0.023 ± 0.001 ^f^	262.5 ± 22.708 ^c^	0.258 ± 0.014 ^e^
0.5 mM Asn	5.225 ± 1.044 ^c^	0.070 ± 0.027 ^d^	287.75 ± 22.248 ^b^	0.726 ± 0.260 ^d^
1 mM Asn	6.103 ± 0.101 ^b^	0.149 ± 0.008 ^b^	316.5 ± 5.196 ^a^	1.375 ± 0.064 ^b^
2 mM Asn	4.673 ± 0.702 ^d^	0.079 ± 0.018 ^c^	314.75 ± 7.411 ^a^	0.846 ± 0.190 ^c^
5 mM Asn	3.428 ± 0.107 ^e^	0.053 ± 0.029 ^e^	256.75 ± 45.339 ^c^	0.383 ± 0.071 ^e^

*P*_n_, Photosynthetic rate; *C*_i_, intercellular CO_2_ concentration; *g*_s_, stomatal conductance; *T*_r_, transpiration rate. Parameters were measured in poplars grown in the hydroponic solution containing different concentrations of Asn as a sole N source. “N+” represents inorganic N control; “N0” indicates N-free control. Different letters behind the values in the same column indicate significant differences between the treatments, tested by one-way ANOVA followed by Dunken’s test (*p* < 0.05).

**Table 3 ijms-23-13126-t003:** Chlorophyll fluorescence parameters of the poplar ‘Nanlin895’ under different asparagine (Asn) treatments.

Treatment	Fv/Fm	NPQ	*Fv*’*/Fm*’	qP	ΦPSⅡ	Fv/Fo
N+	0.783 ± 0.008 ^a^	0.936 ± 0.424 ^ab^	0.66 ± 0.02 ^abc^	0.469 ± 0.057 ^a^	0.307 ± 0.01 ^a^	3.652 ± 0.161 ^a^
N0	0.781 ± 0.003 ^a^	1.065 ± 0.053 ^a^	0.635 ± 0.008 ^c^	0.486 ± 0.018 ^a^	0.308 ± 0.01 ^a^	3.606 ± 0.07 ^a^
0.5 mM Asn	0.773 ± 0.008 ^a^	0.569 ± 0.059 ^bc^	0.688 ± 0.02 ^abc^	0.411 ± 0.014 ^ab^	0.283 ± 0.002 ^bc^	3.478 ± 0.436 ^a^
1 mM Asn	0.79 ± 0.008 ^a^	0.482 ± 0.08 ^bc^	0.717 ± 0.017 ^ab^	0.374 ± 0.014 ^c^	0.268 ± 0.007 ^c^	3.798 ± 0.183 ^a^
2 mM Asn	0.741 ± 0.028 ^b^	0.584 ± 0.171 ^bc^	0.648 ± 0.052 ^bc^	0.433 ± 0.053 ^ab^	0.278 ± 0.012 ^bc^	2.918 ± 0.42 ^b^
5 mM Asn	0.793 ± 0.007 ^a^	0.432 ± 0.38 ^c^	0.73 ± 0.043 ^a^	0.372 ± 0.077 ^c^	0.269 ± 0.037 ^c^	3.877 ± 0.155 ^a^

Fv/Fm, potential maximum light energy conversion efficiency; NPQ, nonphotochemical quenching coefficient; *Fv*’*/Fm*’, Effective quantum efficiency; qP, photochemical quenching coefficient; ΦPSII, actual light energy conversion efficiency; Fv/Fo, photosystem II activity. Different letters behind the values in the same column indicate significant differences between the treatments, tested by one-way ANOVA followed by Dunken’s test (*p* < 0.05).

## Data Availability

Not applicable.

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
