# Peer review of "Effects of Exogenous L-Asparagine on Poplar Biomass Partitioning and Root Morphology"

_ijms, 2022, doi:10.3390/ijms232113126_

Round 1

Reviewer 1 Report

Comments on the ms entitled “Effects of Exogenous L-Asparagine on Poplar Biomass Partitioning and Root Morphology”

The purpose of the work was to evaluate the effects of Asn on the vegetative growth, and physiological and biochemical traits of poplar plants. L-asparagine (Asn) is one of the twenty naturally occurring proteinogenic amino ac ids on the earth. It serves as a principal source of N for protein synthesis, especially in plants’ vigorously growing tissues. L-Asparagine (Asn) has been regarded as one of the most economical molecules for nitrogen (N) storage and transport in plants due to its relatively high N to carbon (C) ratio (2:4) and stability.

The reviewed work provides valuable information and insight on how different forms of N and concentrations of Asn influence poplar root and shoot growth and function, 25 and roles of Asn engaged in protein homeostasis regulation.

My doubts were raised by the markings of the specimens shown in Figures 1a and 1b and the results that were assigned to them in Table 1. Please check this. Details are provided in the text of the manuscript. See lines: 100-101; 121-122 (Fig.1); underline lines in Table 1.

For proofing details of Figure 3, see line 195.

The Authors of this paper presented very well, vividly the schematic model of Asn dose-dependent effects on poplar roots in Fig. 8.

Author Response

My doubts were raised by the markings of the specimens shown in Figures 1a and 1b and the results that were assigned to them in Table 1. Please check this. Details are provided in the text of the manuscript. See lines: 100-101; 121-122 (Fig.1); underline lines in Table 1.

Response: Thank you for your comments. Sorry for our carelessness. The order of Fig 1a and 1b had been swapped. It was revised now.

For proofing details of Figure 3, see line 195.

Response: Thank you for your comments. It has been revised.

The Authors of this paper presented very well, vividly the schematic model of Asn dose-dependent effects on poplar roots in Fig. 8.

Response: Thanks for your encouragement.

Reviewer 2 Report

Amino acids are among the most important primary metabolites in plant cells. The external effects of amino acids affect many physical and chemical characteristics of plant cells and organs. They perform various important biological functions such as detoxification of toxins and heavy metals, resistance to environmental stress, optimization of nutrient absorption, biostimulation of growth. The effect of stimulating the growth of exogenous amino acids is especially high in adverse environmental conditions.

The work was done at a high level and in a large volume.

But it would be interesting to compare the action of exogenous asparagine with some other amino acid (for example, glycine (N-18.8%) or glutamine already studied by the authors). There is no certainty (evidence) that the reactions shown are associated with the maximum presence of nitrogen in asparagine.

In addition, there are some comments:

According to Fig. 1 - the order of photos (a and b) is probably mixed up.

18: poplar or polar?

95: For what reason were these concentrations chosen (0.5, 1, 2, or 5 mM)? Why were such concentrations chosen for treatment?

96: Why was the concentration equal to 3 mM KNO3 used and not other concentrations?

104: Why is there such a spread in values? For example, 22%-645%. Are the data you obtained statistically reliable?

157-159: In the text "the photochemical quenching coefficient (qP) and the actual photochemical quantum efficiency fiction were noticeably lower in Asn-treated plants than that of the N0 and N+ control, suggesting that extra Asn limits photosynthetic capacity. "

However, Table 2 shows that the qP index for 2 mM Asn is lower than for other Asn variants. How can this be explained?

344: There is no closing quotation mark after Nanlin895

358: plants’

Author Response

But it would be interesting to compare the action of exogenous asparagine with some other amino acid (for example, glycine (N-18.8%) or glutamine already studied by the authors). There is no certainty (evidence) that the reactions shown are associated with the maximum presence of nitrogen in asparagine.

Response: Thank you for your comment. Yes, as seen from the available data which have been stated in the text: Line 361-363 and line 337-338 “...the supply of glutamine (0.5 mM) increased NUE in poplar ‘Nanlin895’ [53]. Accordingly, we found that poplar ‘Nanlin895’ feeding with 0.5 mM Asn had higher NUtE and NUpE than the N+ control”; “Our prior study showed that exogenous glutamine as a single N source could promote poplar growth [53]”, both Asn and Gln application had a similar dose-dependent effect on poplar, implying that the reactions shown for Asn treatment for poplar ‘Nanlin895’ might not be associated with the maximum presence of nitrogen in asparagine. Because we are focused on the function and mode of action study for Asn, detailed comparisons among Asn, Gly and Gln were not documented in the current study. It would be interesting to perform this kind of comparison in the future to see whether these amino acids work under a common mechanism or if each had a specific mode of action.

In addition, there are some comments:

According to Fig. 1 - the order of photos (a and b) is probably mixed up.

Response: Thank you for your comments. The order of Fig 1a and 1b had been swapped. It has been revised.

18: poplar or polar?

Response: Thank you for the comment. Yes, it should be poplar.

95: For what reason were these concentrations chosen (0.5, 1, 2, or 5 mM)? Why were such concentrations chosen for treatment?

Response: These concentrations were chosen according to previous literature (Kim, T. H., Kim, E. C., Kim, S. W., Lee, H. S., & Choi, D. W. (2010). Exogenous glutamate inhibits root growth and increases the glutamine content in Arabidopsis thaliana. Journal of Plant Biology, 53(1), 45–51; Haroun, S. A., Shukry, W. M., & El-Sawy, O. (2010). Effect of asparagine or glutamine on growth and metabolic changes in phaseolus vulgaris under in vitro conditions. Bioscience Research, 7(1), 1–21), in which the concentrations used were ranging from 0.05 mM to 5 mM. We performed some pre-tests, omitted those that had a similar effect (i.e., growth performance) to the neighbor concentration, and finally chose 0.5, 1, 2, and 5 for the present study.

96: Why was the concentration equal to 3 mM KNO3 used and not other concentrations?

Response: Previously 4 mM N (2 mM NH4NO3) has been used as regular/intermediate N control for poplars (Cooke, J. E. K., Martin, T. A., & Davis, J. M. (2005). Short-term physiological and developmental responses to nitrogen availability in hybrid poplar. New Phytologist, 167(1), 41–52; Luo, J., Zhou, J., Li, H., Shi, W., Polle, A., Lu, M., Sun, X., & Luo, Z.-B. (2015). Global poplar root and leaf transcriptomes reveal links between growth and stress responses under nitrogen starvation and excess. Tree Physiology, 35(12), 1283–1302). In our case, no significant difference was observed among 2, 3, and 4 mM N for poplar “Nanlin895” growth. As 3 mM N is equal to the concentration of N in the matrix we used in our filed trial for “Nanlin895”, therefore this concentration was employed for the current study.

104: Why is there such a spread in values? For example, 22%-645%. Are the data you obtained statistically reliable?

Response: Thank you for your comments. Sorry for the typo. The initial values were 21.967-64.67%, 62.952-200.999%, 53.571-85.714% and 22.5-131.819%. We round the number of digits. We had wanted to type 22%-65% but accidentally inserted a “4” before 5. It has been revised.

157-159: In the text "the photochemical quenching coefficient (qP) and the actual photochemical quantum efficiency fiction were noticeably lower in Asn-treated plants than that of the N0 and N+ control, suggesting that extra Asn limits photosynthetic capacity. "

However, Table 2 shows that the qP index for 2 mM Asn is lower than for other Asn variants. How can this be explained?

Response: Thanks for the comments. Do you mean “table 3”? Do you mean “the qP index for 2 mM Asn is higher than for other Asn”?

As can be seen from table 3, the qP values for Asn-treated plants were ranging from 0.372 to 0.433, which were significantly lower than the control (0.469 for N+, 0.486 for N0). Despite the qP index for 2 mM Asn being higher than the other Asn concentration, it was still lower than the control. Or in other words, although variants across Asn concentrations occur, such difference is not considerable compared with the difference between the Asn group and the N group.

344: There is no closing quotation mark after Nanlin895

Response: Thanks for the comment. It has been revised

358: plants’

Response: Thanks for the comment. It has been revised